# Plasma Exosomal microRNA Profile Reveals miRNA 148a-3p Downregulation in the Mucosal-Dominant Variant of Pemphigus Vulgaris

**DOI:** 10.3390/ijms241411493

**Published:** 2023-07-15

**Authors:** Anna Valentino, Stefania Leuci, Umberto Galderisi, Gianrico Spagnuolo, Michele Davide Mignogna, Gianfranco Peluso, Anna Calarco

**Affiliations:** 1Research Institute on Terrestrial Ecosystems (IRET)—CNR, Via Pietro Castellino 111, 80131 Naples, Italy; anna.valentino@cnr.it (A.V.); anna.calarco@cnr.it (A.C.); 2National Biodiversity Future Center (NBFC), 90133 Palermo, Italy; 3Oral Medicine Unit, Department of Neurosciences, Reproductive and Odontostomatological Sciences, Federico II University of Naples, 80138 Naples, Italy; stefania.leuci@unina.it (S.L.); gspagnuo@unina.it (G.S.); mignogna@unina.it (M.D.M.); 4Department of Experimental Medicine, University of Campania “Luigi Vanvitelli”, Via Santa Maria di Costantinopoli, 80100 Naples, Italy; umberto.galderisi@unicampania.it; 5Faculty of Medicine and Surgery, Saint Camillus International University of Health Sciences, Via di Sant’Alessandro 8, 00131 Rome, Italy

**Keywords:** circulating exosomes, microRNAs, mucosal lesions, metalloprotease, pemphigus vulgaris

## Abstract

The mucosal-dominant variant of pemphigus vulgaris (MPV) is an autoimmune disease characterized by oral mucosal blistering and circulating pathogenic IgG antibodies against desmoglein 3 (Dsg3), resulting in life-threatening bullae and erosion formation. Recently, microRNAs (miRNAs) have emerged as promising players in the diagnosis and prognosis of several pathological states. For the first time, we have identified a different expression profile of miRNAs isolated from plasma-derived exosomes (P-EVs) of MPV patients positive for antibodies against Dsg3 (Dsg3-positive) compared to healthy controls. Moreover, a dysregulated miRNA profile was confirmed in MPV tissue biopsies. In particular, a strong downregulation of the miR-148a-3p expression level in P-EVs of MPV patients compared to healthy controls was demonstrated. Bioinformatics prediction analysis identifies metalloproteinase-7 (MMP7) as a potential miR-148a-3p target. An in vitro acantholysis model revealed that the miR-148a-3p expression level was dramatically downregulated after treatment with Dsg3 autoantibodies, with a concomitant increase in MMP7 expression. The increased expression of MMP7 leads to the disruption of intercellular and/or extracellular matrix adhesion in an in vitro cellular model of MPV, with subsequent cell dissociation. Overexpression of miR-148a-3p prevented cell dissociation and regressed MMP7 upregulation. Our findings suggest a pivotal role of P-EV cargo in regulating molecular mechanisms involved in MPV pathogenesis and indicate them as potential MPV therapeutic targets.

## 1. Introduction

Pemphigus vulgaris (PV) is a chronic, potentially lethal autoimmune mucocutaneous bullous disease mediated by essentially IgG autoantibodies directed against desmogleins 1 and 3 (Dsg1 and Dsg3), cadherin-type cell–cell adhesion molecules present in desmosomes [1]. 

Because of the rarity of the disease, patients must wait many weeks or even months until diagnosis, during which period they can be mistakenly treated for other blistering diseases. Diagnosis is established based on clinical presentation, characteristic immunohistopathology with suprabasal detachment and perilesional deposition of IgG and/or C3, and serologic detection of autoantibodies against Dsg3 and/or Dsg1 epithelial cell surface antigens [2]. Even once the diagnosis has been made, there is still a need to control the disease activity with measurable bio-parameters (biomarkers) to be used for monitoring disease progression, prognosis, and drug response, thus optimizing the choice of appropriate and often personalized therapies [3]. 

Although several factors have been proposed as markers of disease duration and relapse risk in PV, studies have been inconclusive, and their clinical use is not yet recommended [4]. Indeed, the identification of specific biomarkers is often made difficult by both the low number of cases available and the clinical effects of previously prescribed treatment, making a sound evaluation of sensitivity, specificity, and predictive value difficult. Indeed, pharmacological treatments with dexamethasone have shown that the secretion pathway of target cells can change quickly after the beginning of the therapy with corticosteroids [5]. Recently, microRNAs (miRNAs) have been proposed as a novel option for predicting the outcome of PV. MiRNAs represent a family of small, non-coding RNAs involved in the post-transcriptional regulation of gene expression. Numerous papers have reported that several miRNAs (miR-424-5p, miR-338-3p, and miR-584-5p) are aberrantly expressed in the peripheral blood mononuclear cells (PBMCs) of pemphigus patients [6,7]. Studies on lesional keratinocytes obtained from patients affected by familial benign chronic pemphigus (Hailey–Hailey disease, HDD) have shown a specific pattern of miRNA expression, characterized by upregulation of miR-125b, miR-181, and miR-99, and downregulation of miR106a, miR106b, and miR-18b [8]. 

However, although emerging evidence underscores miRNAs as potential novel biomarkers of pemphigus diagnosis, severity, and prognosis assessment, miRNA research in the mucosal-dominant variant of pemphigus (MPV) is still in its infancy. Indeed, the majority of the studies have mainly focused on pemphigus foliaceus, paraneoplastic pemphigus, HHD, or pemphigoid diseases [9,10,11]. Only recently did a scientific paper report that twelve miRNAs were increased in the serum of MPV patients compared with control subjects, in which six miRNAs (including miR-125b-5p, miR-155-5p, miR-181a-5p, miR181b-5p, miR-326, and miR-584-5p) demonstrated high diagnostic values for MPV [12]. 

Our experimental approach has been quite different from other previous studies since we have analyzed, in MPV patients, plasma-derived exosomes (P-EVs) as miRNA carriers. The aim of this study was to investigate P-EV-incorporated microRNA profiles that are dysregulated in patients with MPV compared to healthy subjects. The correlations between the P-EV-miRNA expression pattern and miRNA expressed in biopsies from affected and unaffected mucosa were also evaluated. Finally, target gene analyses and an in vitro study on the role of Dsg autoantibodies in inducing miRNA dysregulation were performed. In summary, our study identified a plasma exosomal miRNA profile associated with MPV. Again, we demonstrated the central role played by miR-148a-3p, the decrease of which is connected to a concomitant increase in metalloproteinase-7 (MMP7) protein, an enzyme able to cleave the extracellular domain of both Dsgs and other cell adhesion molecules.

## 2. Results

### 2.1. Patient Selection

Patients were referred to the Oral Medicine Unit, Department of Neuroscience, Reproductive and Odontostomatological Sciences, Federico II University of Naples, with suspected MPV. The patients underwent a complete clinical examination, laboratory tests, including an ELISA test to detect antibodies anti-Dsg1 and anti-Dsg3, and an incisional oral biopsy with direct immunofluorescence (DIF). Five patients (three females and two males) met eligibility criteria and were enrolled in the study. The mean age was 56.6 years (range: 36–67 years). The controls were three males and two females, with a mean age of 55.4 years (range: 45–61 years). In the case group, clinical inspection showed blisters and erosions involving the oral mucosa (Figure 1A–C). On histopathological examination, all MPV patient samples showed acantholysis with intraepidermal blister formation (Figure 1D,E). DIF demonstrated the presence of intraepidermal deposits of IgG and/or C3 with a “fishnet” intercellular pattern. The results were corroborated by immunofluorescence analysis (Figure 1F,G). 

### 2.2. Dsg3-Positive Exosomes in Peripheral Blood

P-EVs were characterized and quantified via nanoparticle tracking analysis (NTA), dynamic light scattering (DLS), and Western blotting. Neither the mean size (110 ± 5 nm) nor the plasma concentration (5.45 ± 0.73 × 10^9^ particle/mL) of P-EVs were affected in MPV patients in comparison to controls (Figure 2A,B). As expected, Western blot analysis demonstrated signals for the typical P-EV markers, CD63 and TSG101, in P-EV-derived proteins of both, healthy volunteers and MPV patients [13] (Figure 2C). 

Previous data reported that human Dsg3 is expressed in MPV lesions [14]. Thus, we analyzed P-EV by fluorescent NTA analysis using the fluorescence-conjugated Dsg3 antibody. Indeed, while conventional NTA allows the detection of total exosomes, the use of a fluorescent membrane marker permits the phenotypical characterization of distinct subsets of exosomes within a sample. Figure 3 depicts that no difference between MPV and control P-EVs was found in terms of CD63 expression (Figure 3B), while the percentage of Dsg3-expressing exosomes was significantly higher (*p* < 0.01) in MPV plasma samples compared to controls (70% vs. 8%; Figure 3D). As reported in Appendix A, Western blot analysis confirmed higher Dsg-3 expression in MPV exosomal protein with respect to P-EVs obtained from healthy subjects.

### 2.3. Exosomal miRNA Expression Profile and miR-148a-3p Study

miRNA-Seq analysis was performed on P-EVs from both groups (MPV and healthy subjects). 

The total number of P-EV miRNAs detected was approximately 734, and almost 89 of these transcripts were shared between MPV and control P-EVs. After raw data normalization, 25 out of 89 miRNAs were differentially expressed between the two groups (Appendix A). Specifically, miR-424-5p showed higher expression in MPV patients compared to controls, with a statistical significance of *p* < 0.05. Moreover, miR-146b-5p, miR-126, and miR-139 were found to be lower in MPV patients compared to healthy subjects (*p* < 0.05), while a marked downregulation was evident for miR-148a-3p (*p* < 0.001).

The data were corroborated by Taqman PCR results. As shown in Figure 4A, miR-424-5p was the most abundant miRNA species in P-EVs from MPV patients, while miR-148a-3p, miR-146b-5p, miR-126, and miR-139 were significantly lower in MPV patients than in healthy subjects. Of interest, the analysis of RNA on MPV biopsies concluded that the miRNA profile, relative to the five miRNAs that reached the threshold of a ≥1.5-fold change in MPV vs. control exosomes, also followed the same trend in tissue samples and PV cell model (Appendix A). 

A miRNA enrichment analysis for miR-424-5p, miR-146b-5p, miR-126, and miR-139 revealed that the present collection of MPV exosome miRNAs was significantly associated with molecular functions related to the MAPK pathway [6], invasion and cell metastasis via Wnt/β-catenin signaling [15], angiogenesis and proliferation via the PI3K/AKT signaling pathway [16], and senescence and apoptosis, respectively [17]. 

Considering that miR-148a-3p is constantly downregulated in both MPV exosomes and injured tissue biopsies, with the largest difference between MPV and control samples, a comprehensive network of miR148a-3p functional pathways was obtained using miRPath v 2.0 (https://mpd.bioinf.uni-sb.de/ (accessed on 2 March 2022)). As depicted in Figure 4B, miR148a-3p may modulate the translation of many factors involved in extracellular matrix remodeling also connected to each other (Appendix A).

A subsequent assessment by qPCR on the expression of miR-148a-3p potential targets demonstrated a significant increase in transforming growth factor-β (*TGF-β*) (*p* < 0.05) and matrix metalloproteinase 7 (*MMP-7*) (*p* < 0.001) in MPV biopsies with respect to CTL. The qPCR results were confirmed by Western blot analysis (Figure 4D). To verify if MMP7 is functionally active, we tested one of its potential substrates, E-cadherin [18,19]. As shown in Figure 4E,F, the E-cadherin gene expression and protein level were downregulated in MPV patient biopsies compared to healthy controls.

### 2.4. MPV IgG Induced MMP7 via miR-148a-3p Downregulation in Primary Human Keratinocytes

The use of Dsg3 autoantibodies derived from the sera of PV patients, both in in vivo and in vitro PV models, has demonstrated the crucial role played by these antibodies in inducing classical PV hallmarks, such as widening of intercellular spaces and disrupted desmosomes with loss of intercellular adhesion [20]. 

Based on these studies, primary human keratinocytes (HEKa) were incubated in the presence of IgG fractions derived from PV patient sera (MPV-IgG) tested for the presence of specific autoantibodies against Dsg3. As shown in Figure 5A, MPV-IgG fractions caused keratinocyte dissociation with loss and disruption of desmosome aggregation due to Dsg3 shift from the membrane to the cytoplasm. Dsg3 immunostaining demonstrated Dsg3 fragmentation along the cell borders and its internalization into the cell (Figure 5B). The loss of contact between neighboring cells and disruption of adherent junctions were also corroborated by E-cadherin staining (Figure 5B) [21,22]. On the contrary, cells incubated in the presence of IgG obtained from healthy volunteers (c-IgG) remained unchanged for all the culture period, with Dsg3 linearly organized at cell borders (Figure 5B). 

Besides loss of cell–cell adhesion, other in vitro readouts have been used in our PV cell model, which may be causative for pathogenicity. Indeed, binding of anti-Dsg3 autoantibodies to extradesmosomal Dsg3 seems to be linked to outside-in signal transmission due to a membrane-receptor-like function for non-desmosomal Dsg3 [23]. 

Due to the central role of miR-148a-3p in modulating the expression of proteins involved in extracellular matrix remodeling, we verified if miRNA expression was affected in our in vitro PV model. As reported in Figure 5C, the miR-148a-3p level is significantly downregulated in MPV-IgG-treated cells with respect to c-IgG control cells. Moreover, transfection with miR-148a-3p mimics led to miRNA overexpression, while treatment with a miRNA inhibitor did not cause any miRNA level variation.

Then, we analyzed the expression of potential target of miR-148a-3p, as shown in Figure 5D, E.

After mimic transfection, MMP7 gene expression was significantly downregulated in MPV-IgG transfected cells with respect to no-transfected cells (*p* < 0.01) (Figure 5D). The MMP7 protein level, determined by Western blotting, was consistent with qPCR results (*p* < 0.01) (Figure 5E). In addition, following miR-148a-3p mimic transfection, a reduction in fragment number was demonstrated, as shown in Figure 5A, and a recovery of the cell joints can be noted (Figure 5B).

## 3. Discussion

Membrane vesicles (MVs) are secreted by all cell types, both prokaryotes and eukaryotes, even though the number, size distribution, and content depend on the state of the cell [24,25]. Indeed, MVs, and in particular exosomes, the smallest MVs with diameters ranging from 30–150 nm, have gained considerable scientific interest in cell–cell communication due to their ability to transfer functional proteins, metabolites, and nucleic acids to recipient cells [26,27]. 

In addition, P-EVs have also been studied as diagnostic platforms for multiplexed approaches. Several scientific papers have reported disease-specific miRNA signatures in exosomes obtained from patients with a wide range of diseases and conditions. In the case of cancer patients, the exosomal miRNA profiling content has made it possible to discriminate malignant from benign disease [28,29]. Of interest, a retrospective study has shown that the levels of individual miRNAs were not affected by storage time and that the sensitivity of this approach for cancer diagnosis was significantly higher when compared with other serum cancer biomarkers [30]. Indeed, because P-EVs come from multivesicular bodies protected from RNases, miRNAs within exosomes have longer life spans [31,32].

Recently, it has been demonstrated that P-EVs may also be ideal biomarkers for diagnosing oral diseases, such as oral lichen planus, hand-foot-and-mouth disease, or Sjogren’s syndrome. P-EV miRNA profiles differ between patients suffering from diseases of the oral cavity and healthy controls, suggesting that either pathophysiological changes can selectively modulate the miRNA loading mechanism or the P-EV miRNA subset mirrors those changes present in the cells they originate from [33,34].

For the first time, we have demonstrated that MPV patients showed an increase in circulating Dsg3^+^ P-EVs with a specific miRNA cargo, which appears to be characteristic of the MPV disease process. Indeed, we identified five miRNAs with a significantly altered expression level in P-EV-derived MPV patients. Of interest, this dysregulated miRNA profile was also confirmed in tissue biopsies obtained from oral lesions in MPV patients. A pathway enrichment analysis revealed that the selected miRNAs are involved in a range of pathways, including invasion and cell metastasis, angiogenesis and proliferation, senescence, apoptosis, and matrix remodeling. These pathways play critical roles in the regulation of different biological processes and are often interconnected, highlighting the complex regulatory network monitored by miRNAs in MPVs. In particular, the dysregulation of miR-148a-3p affects the expression of enzymes, such as metalloproteinase 7, that are able to degrade extracellular matrix proteins and extracellular portions of transmembrane proteins. Metalloproteinases are secreted by both keratinocytes and infiltrating immune cells and are good candidates for mediating the disruption of intercellular and/or cell-extracellular matrix adhesion. However, despite the well-known ability of metalloproteinases to cleave the extracellular domain of both Dsgs and other cell adhesion molecules [35,36], data relating to their involvement in the pathogenesis of MPV are scarce. Interestingly, in our in vitro experiments, the miR-148a-3p expression level was dramatically downregulated with a concomitant increase in MMP7 after cell culture exposure to MPV-inducing autoantibodies. The increased expression of MMP7 decisively influenced acantolysis, since cell transfection with miR-148a-3p mimic induced a significant decrease in MMP7 that prevented cell dissociation. Despite the relatively limited number of studies investigating the role of metalloproteinases in autoimmune blistering diseases, inactivation of these enzymes was shown to prevent blister formation in experimental models of PV induced by PV IgG autoantibodies [37].

Our findings provide insights into molecular mechanisms and potential targets for treating MPV. As outlined in the study, a limitation of this research is the number of patients enrolled, although the results on miRNA deregulation were also confirmed in an in vitro cell model of MPV. Further studies involving larger groups and long-term monitoring are required to confirm the results and discover several other miRNAs as modulators of cell adhesion molecules, thus founding new layers in the intricate pathogenesis of MPV.

## 4. Materials and Methods

### 4.1. Patients, Study Design, and Approvals

This study was conducted according to the principles expressed in the Declaration of Helsinki. Ethical approval for this study was obtained from the Ethics Committee of the Federico II University of Naples (N°69/19). Written informed consent was obtained from all study participants before they participated in this study. Adult patients (aged 18–75 years) of both genders who met the above diagnostic criteria were included in the MPV group as determined by experts in oral medicine. Normal controls (CTL) were adults without systemic diseases or medication histories (Table 1). The inclusion criteria were as follows: (a) newly diagnosed bullous and/or erosive lesions affecting the oropharyngeal mucosa with or without other mucosal surfaces or skin involvement at the time of diagnosis; (b) histopathological features of PV with intra-epithelial detachment; (c) deposition of IgG complement component 3, or both, on the keratinocyte membrane detected by direct immunofluorescence and/or evidence of antibody anti-Dsg3 detected by enzyme-linked immunosorbent assay. The patients were first subjected to a complete clinical interview and a clinical examination, including a manual and visual inspection of intra- and extra-oral sites. An incisional biopsy, including the lesional and perilesional areas, was performed by an oral medicine specialist with a routine histological examination by a pathologist at the same university. Moreover, DIF was performed to detect deposits of Igs-G, Igs-A, Igs-M, C3, and fibrinogen, and an ELISA test was made to detect antibodies Dsg3, BP-180, and BP-230.

### 4.2. Sample Collection

Tissue biopsies and peripheral blood samples from MPV patients (MPV) and the CTL group were used for P-EV isolation, microRNA/RNA, and protein extraction. 

In particular, around 10 mL of venous blood samples were taken from the cubital vein and collected in ethylenediaminetetraacetic acid (EDTA) tubes. Plasma was retained by centrifugation at room temperature for 10 min at 1000× *g* (Frontiers 5718R, OHAUS, Milan, Italy). The resulting supernatants were transferred to a clean tube and centrifuged again at 3000× *g* for 15 min before being aliquoted, and stored at −80 °C until further processing. 

Tissue samples were sliced into small pieces with a scalpel and processed as reported below for microRNA/mRNA and total protein extraction. 

### 4.3. MPV-IgG Purification

Patients’ autoantibody profiles were tested by enzyme-linked immunosorbent assay (ELISA, Sigma-Aldrich, Milan, Italy) according to the manufacturer’s protocols for reactivity against Dsg3. IgG fractions from MPV patients (MPV-IgG) and the CTL group (c-IgG) were purified by affinity chromatography using protein A agarose (Sigma-Aldrich, Milan, Italy) as described by Waschke et al. [38,39]. The purity of isolated IgGs was checked by Coomassie blue staining of 7.5% SDS-PAGE. The concentrations of IgG fractions were adjusted to a final concentration of 150 μg/mL for all experiments.

### 4.4. In Vitro MPV Cell Model

Primary human epidermal keratinocytes (HEKa) were obtained from ThermoFisher Scientific (Roma, Italy) and maintained in complete medium (Dulbecco’s Modified Eagle Medium (DMEM)) containing 10% heat-inactivated fetal bovine serum, penicillin (100 U/mL), and streptomycin (100 mg/mL) at 37 ℃ in a humidified atmosphere with 5% CO_2_ [40]. 

To simulate PV disease, cells (70% confluency) were cultured in low calcium media (50 mM CaCl_2_) overnight to internalize all desmosomal cadherins. Then, cultures were switched to high calcium (550 mM CaCl_2_) media for 3 h and treated with MPV-IgG or c-IgG for 24 h. After incubation with IgGs for the indicated times, HEKa cells were used for total protein and mRNA isolation as reported in Sections Section 4.10 and Section 4.11.

For the Keratinocyte Dissociation Assay, cells were incubated with dispase (Sigma-Aldrich, Milan, Italy) for 15 min to release cells as monolayers, as reported by Cho et al. [41].

For immunofluorescence experiments, cells were cultured directly on glass slides until confluent, as described above. Cells were fixed on ice using 3.7% paraformaldehyde for 10 min, followed by permeabilization in 0.1% triton x-100 for 10 min at room temperature, and labeled with monoclonal antibodies (anti-dsg3 or anti-E-cadherin, Cell Signaling, Milan, Italy; dilution 1:100 in PBS) at a concentration of 10 mg/mL for 1 h at 4 °C. After several rinses with phosphate-buffered saline (PBS), monolayers were incubated for 60 min at room temperature with Cy5-labeled goat anti-mouse IgG (Cell Signaling) prior to mounting with prolong gold-containing Dapi (Thermofisher). Widefield images were acquired using a Vert.A1 microscope (Zeiss, Oberkochen, Germany).

### 4.5. P-EV Isolation and Purification

Plasma samples were differentially centrifuged at 300× *g* for 10 min and 2000× *g* for 20 min to pellet cell fragments and other debris. All samples were centrifuged at 10,000× *g* × 30 min at 4 °C in QuickSeal^®^ Polypropylene tubes (Beckman Coulter, Fullerton, CA, USA Ref. 342414) to remove large particles (diameter of about 1000 nm) and then ultracentrifuged twice at 120,000× *g* for 2 h at 4 °C in Beckman Coulter Optima L-100 XP ultracentrifuge (Fullerton, CA, USA) equipped with a fixed-angle Type 70 Ti rotor. P-EVs were resuspended in filtered PBS, aliquoted in samples of 1 × 10^9^, and preserved at −80 °C.

### 4.6. EV Characterization

#### 4.6.1. Dynamic Light Scattering (DLS)

The size distribution of P-EVs was analyzed by DLS (Zetasizer Ultra, Malvern Panalytical, Amesbury, UK). The DLS technique analyzes the velocity distribution of particle motion caused by Brownian motion by measuring dynamic fluctuations in the intensity of scattered light. The hydrodynamic radius of the particle, or diameter considered, is calculated with the Stokes–Einstein equation. Ten microliters of purified P-EVs were diluted in 990 μL of filtered PBS and vortexed. To avoid aggregation of exosomes, the entire volume was quickly placed into a disposable cuvette for size measurements. Three independent measurements were performed for each sample, and the analysis was processed by the software.

#### 4.6.2. Nanoparticle Tracking Analysis

Nanoparticle tracking analysis (NTA) was used to analyze the size and concentration of P-EVs. Briefly, samples were diluted with 0.1 μm-filtered PBS to obtain a recommended measurement concentration between ~10^8^–10^9^ particles/mL. All NTA measurements were carried out at a frame rate ≥ of 25 frames/s using a NanoSight NS300 (Malvern Panalytical Ltd., Malvern, Worcestershire, UK) with a 488 nm laser. The temperature was maintained at 25 °C. The data were analyzed using NanoSight NTA software version 3.2. To validate the operation of the instrumentation, polystyrene beads (Malvern, Spherotech) of 100 nm and/or 200 nm were used as standards. Filtered PBS (blank) was run as a negative control. 

### 4.7. P-EV Enrichment Using Magnetic Beads

For detecting exosome-associated surface proteins, P-EVs were enriched using magnetic beads according to the manufacturer’s protocol (ThermoFisher Scientific). In brief, anti-CD63 or anti-Dsg3 antibody-conjugated magnetic beads (100 μL, bead diameter 4.5 μm, 1 × 10^7^ beads/mL) were mixed with 100 μL P-EVs and incubated at 4 °C for 18 h. P-EVs captured on magnetic beads were concentrated on the magnetic rack and washed twice with the isolation buffer. The captured exosomes were labeled with a fluorescent Vybrant™ DiO dye solution (5 µL/mL, Molecular Probes, Life Technologies, Milan, Italy) for 8 min at 37 °C to allow staining of the exosomal membrane. Labeled P-EVs were rinsed with PBS and analyzed by Fluorescent NTA (fNTA, Malvern, UK) equipped with a 488 nm laser and a 500 nm long-pass filter for fluorescence detection. For fNTA analysis, samples were diluted to provide a concentration of 1 × 10^8^–1 × 10^9^ particles/mL. Each sample was performed in five replicates, recording a 30–60 s video with a minimum of 200 valid traces (a minimum of 1000 valid traces were recorded per sample). Data were analyzed by Nanosight NTA version 3.2 Analytical Software (Malvern), with the detection threshold optimized for each sample and screen gain set at 10 to track as many particles as possible with minimal background. All camera and detection threshold settings were kept the same for each mode when performing multiple experiments to minimize variability. Moreover, all immunolabeled samples were evaluated first in fluorescence mode to minimize photobleaching, followed immediately by evaluation in light scatter mode.

### 4.8. Exosomal Small RNA-Seq Analysis

Total exosomal RNA was extracted from EVs using a ExoRNeasy mini-kit (Qiagen, Milan, Italy) according to the manufacturer’s instructions. Identification and quantification of miRNAs were undertaken with Small RNA-Seq (Sequentia Biotech SL, Barcelona, Spain). In brief, Illumina NGS libraries were prepared and sequenced using an Illumina HiSeq2000 sequence analyzer (Illumina Inc., San Diego, CA, USA). Before further analysis, a quality check was performed on the raw sequencing data, deleting low quality portions while preserving the longest high-quality part of NGS reads. The high-quality reads were aligned against the Homo sapiens genome (Ensembl GRCh38) with a STAR aligner (version 2.7.7a). STAR can align spliced sequences of any length with moderate error rates, providing scalability for emerging sequencing technologies. STAR generates output files that can be used for many downstream analyses, such as transcript/gene expression quantification. Out of the 734 miRNAs present in the official annotation, a differential expression analysis between the groups (MPV vs. CTL) was performed. 89 miRNAs were identified, 25 of them were dysregulated (2 upregulated and 23 downregulated). 

### 4.9. MiRNA Target Predictions

The miRNA predicted targets analysis was carried out using the algorithms miRBase (http://www.mirbase.org/ (accessed on 2 March 2022)), miRDB (http://mirdb.org/miRDB/ (accessed on 2 March 2022), miRTar (http://mirtar.mbc.nctu.edu.tw/human/ (accessed on 2 March 2022)), DIANA Lab (http://diana.cslab.ece.ntua.gr/ (accessed on 2 March 2022), and TargetScan (http://www.targetscan.org/ (accessed on 2 March 2022). The algorithm produced a list of hundreds of target genes for selected miRNAs by searching for the presence of conserved 8-mer and 7-mer sites matching the miRNA “seed region,” miRNA binding site conservation, and energy of binding to miRNA. Moreover, with the help of some software, such as MiRPath v 2.0, a network of putative targets of miRNA was found to be deregulated in patients compared to controls.

### 4.10. Quantitative Real-Time PCR (qPCR)

Total RNA was extracted using QIAzol reagent (QIAGEN, Milan, Italy), according to the manufacturer’s protocols, and reverse transcribed as reported in Di Cristo et al. [42]. qPCR was performed in a 7900HT Fast Real time-PCR System (Applied Biosystems, Milan, Italy) with SYBR Green PCR Master Mix (Invitrogen, Milan, Italy) according to the manufacturer’s instructions. All reactions were run in triplicate, normalized to the housekeeping gene (*ACTIN*), and analyzed using the 2^−∆∆Cq^ method. The primers used in the qPCR are listed in Table 2.

qPCR for selected miRNA detection was performed using TaqMan MicroRNA Assays (Applied Biosystems) following the manufacturer’s instructions. The expression *RNU6B* was used for quantitative normalization. All experiments were performed in triplicate, and the results were expressed as mean ± SD. Specific miRNA TaqMan probes for hsa-miR-146b-5p, hsa-miR-126, hsa-miR-139, hsa-miR-148a-3p, and hsa-miR-424-5p were purchased from Applied Biosystems. 

### 4.11. Protein Extraction and Western Blot Analysis

Total and exosome proteins were isolated using RIPA buffer (Sigma-Aldrich) according to the manufacturer’s instructions. The protein concentration was obtained by the Bradford assay following the manufacturer’s protocol. From each prepared sample, 30 µg of lysed protein was separated using 8% sodium dodecyl sulphate-polyacrylamide gel (SDS-PAGE) and blotted onto Immobilon-P polyvinylidenedifluoride membranes (MPVDF, Millipore, Milan, Italy). Membranes were blocked with 5% skim milk and then incubated with primary antibodies detecting anti-MMP7, anti-TGFβ, anti-E-cadherin, CD-63, TSG101, anti-Dsg3, and anti ACTIN (Cell Signaling), followed by secondary antibodies conjugated with horseradish peroxidase. ImageJ 1.8.0 (National Institutes of Health, Bethesda, MD, USA) was applied to quantify the relative protein levels.

### 4.12. Statistical Analysis

R package DESeq version 3.1.1 available in Bioconductor (Release 3.0) was considered to determine the statistical significance of differential expression measured by miRNA sequencing. DESeq provides a method to identify differentially expressed miRNAs using the negative binomial distribution with variance. All *p*-values obtained were adjusted for the false discovery rate (FDR < 0.05). All quantitative data were presented as the mean ± SD. Each experiment was performed at least six times. Statistical significance was evaluated using a *t*-test or one-way analysis of variance, followed by Bonferroni’s test for multiple comparisons to determine statistical differences between groups. All the data were analyzed with the GraphPad Prism version 5.01 statistical software package (GraphPad, La Jolla, CA, USA).

## Figures and Tables

**Figure 1 ijms-24-11493-f001:**
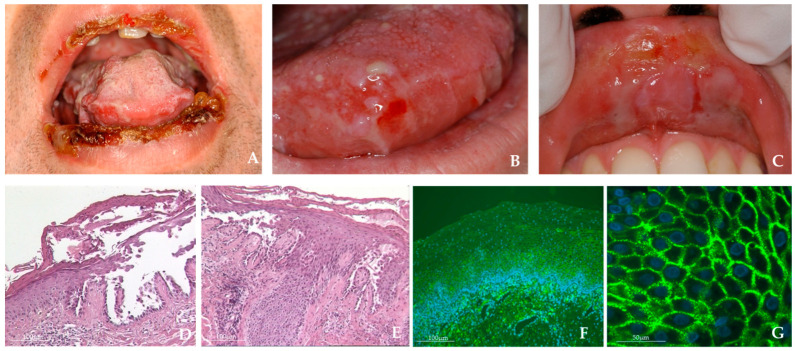
(**A**) 50-year-old male; crusted erosions and blistering lesions in the lower and upper lip. (**B**) 36-year-old female; blistering lesions of the tongue. (**C**) 36-year-old female; erosive and bullous lesions of the upper labial mucosa (the white aspect represents the partial collapsed roof of the bulla). (**D**,**E**) Representative images of histopathological examination from PV patients showing suprabasal bulla with acantholysis (hematoxylin-eosin, 40×). (**F**,**G**) Direct immunofluorescence microscopy with typical staining for Dgs 3, with an intercellular fluorescence distribution of 10× (**F**) and 40× (**G**).

**Figure 2 ijms-24-11493-f002:**
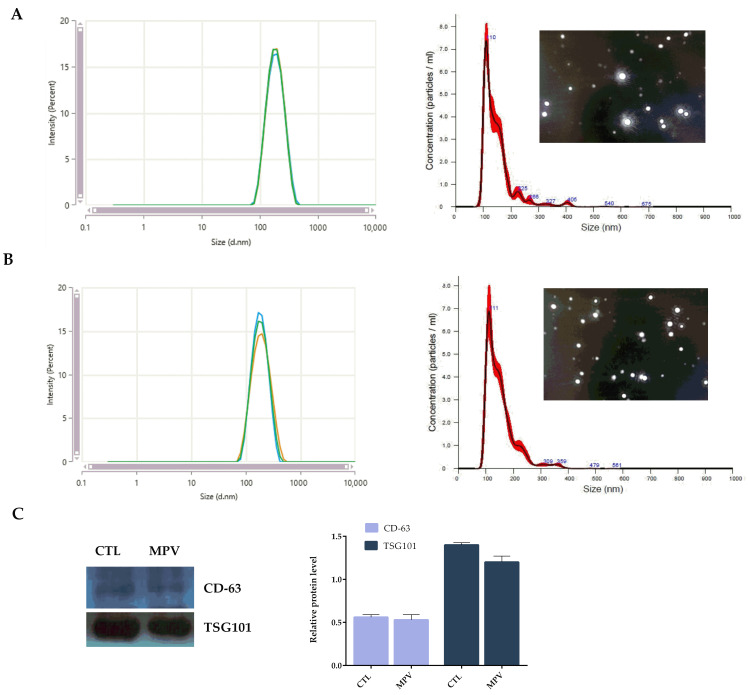
Size distribution plots from dynamic light scattering (DLS), nanoparticle tracking analysis (NTA), and screenshots of representative NTA videos of P-EVs isolated from healthy subjects (**A**) and MPV patients (**B**). Hydrodynamic diameter distribution curves are the average of three measurements. (**C**) Western blot analysis and relative quantification of protein expression of P-EVs-associated tetraspanin CD63 and cytosolic endosomal sorting complex component TSG101 of P-EVs isolates from healthy subjects and MPV patients. The protein levels were quantified using ImageJ.

**Figure 3 ijms-24-11493-f003:**
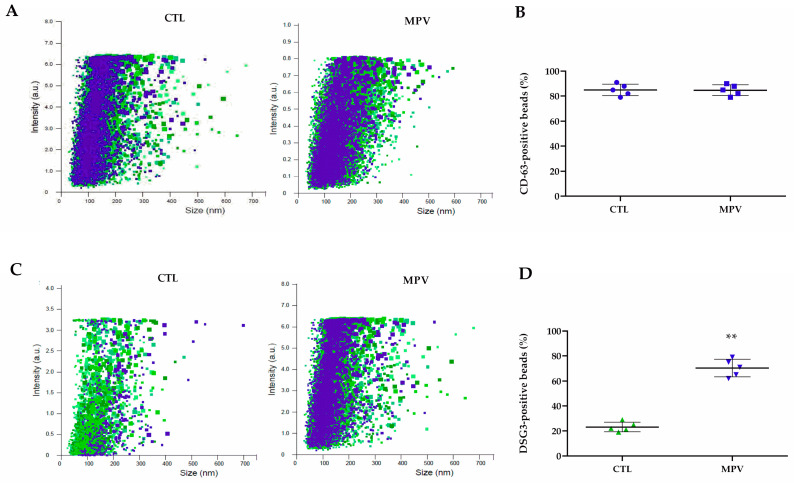
Comparison of exosome surface marker expression in MPV and healthy (CTL) P-EVs using fNTA. Fluorescence intensity of (**A**) CD63-labeled P-EVs and (**C**) Dsg3-labeled P-EVs derived from healthy and MPV samples. Fluorescence intensity quantification of (**B**) CD63-labeled P-EVs and (**D**) Dsg3-labeled P-EVs (detected particle concentration: 1 × 10^9^ particles/mL). The color represents the signal intensity, low (green) high (blue). The bars represent the mean ± S.D (n = 6). Statistically significant variations: ** *p* < 0.01, versus CTL.

**Figure 4 ijms-24-11493-f004:**
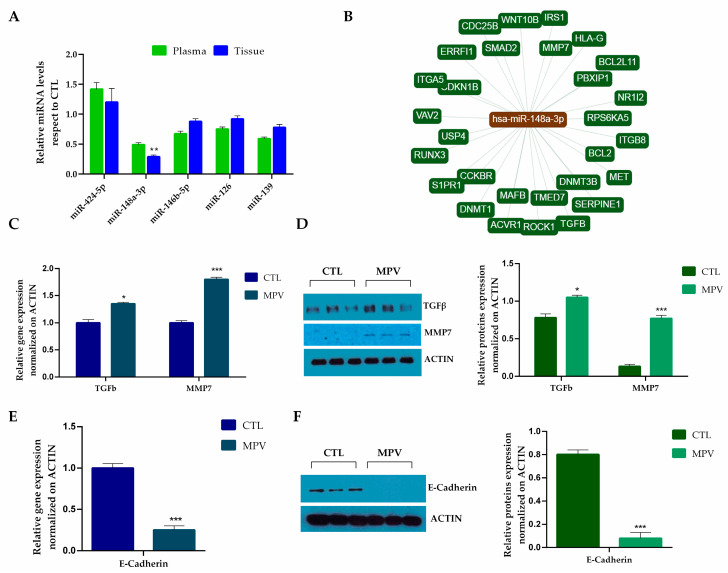
(**A**) Fold change of the five miRNAs in both plasma and tissue samples in MPV patients with respect to healthy controls determined by the TaqMan-RT-PCR assay. (**B**) miR-148a-3p target network realized using miRPath. (**C**) *TGF-β* and *MMP7* gene expression analysis in MPV tissue samples with respect to healthy subjects determined by RT-qPCR. Relative gene expression was normalized vs. ACTIN. (**D**) Western blot analysis of TGF-β, and MMP7 total protein expression in MPV tissue samples respect to healthy subjects. (**E**) *E-cadherin* gene expression analysis in MPV tissue samples with respect to healthy subjects determined by RT-qPCR. (**F**) Western blot analysis of E-cadherin total protein expression in MPV tissue samples with respect to healthy subjects. The comparative cycle threshold (CT) method (2^−ΔΔCT^) was applied to calculate relative differences. ACTIN was used as the control protein. The protein levels were quantified using ImageJ. The bars represent the mean ± S.D. (n = 6). Statistically significant variations * *p* < 0.05, ** *p* < 0.01, *** *p* < 0.001 versus CTL.

**Figure 5 ijms-24-11493-f005:**
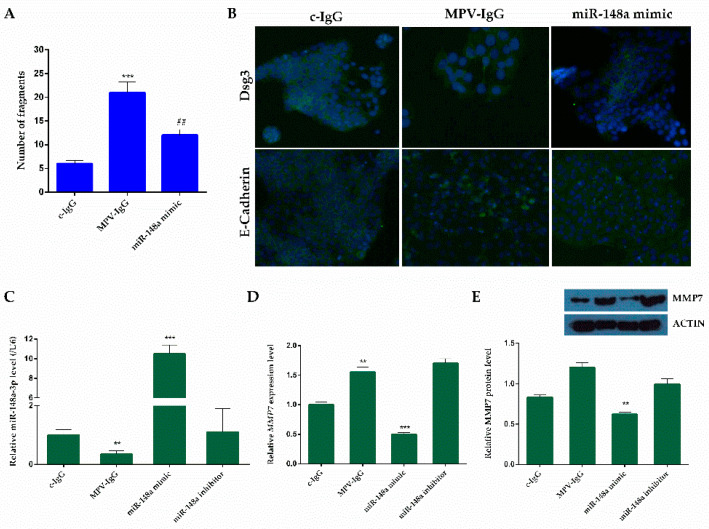
Pemphigus autoantibody application induces loss of keratinocyte cohesion and fragmentation of Dsg3 immunostaining. (**A**) Dispase-based dissociation assay in HEKa cells after treatment with MPV-IgG and after miR-148a-3p mimic transfection. Cells exposed to IgG obtained from healthy volunteers (c-IgG) were used as a control. The bars represent the mean ± S.D (n = 6). Statistically significant variations: *** *p* < 0.001 versus c-IgG, ## *p* < 0.01 versus MPV-IgG. (**B**) Dsg3 and E-Cadherin immunostainings (green) in response to MPV-IgG and miR-148a-3p mimics; c-IgG was used as a control. Nuclei were stained with DAPI (blu). Representative images of n ≥ 4 (scale bar: 200 μm). (**C**) qPCR validation of expressed miR-148a-3p in HEKa transfected with mimics with respect to the inhibitor. The detection of miRNA was performed by the TaqMan qPCR miRNA assay and normalized to RNU6B. The comparative cycle threshold (CT) method (2^−ΔΔCT^) was used to calculate relative differences in PCR data. (**D**) Validation of MMP-7 as a target of miR-148a-3p in HEKa cells; the *MMP7* level was detected in HEKa cells by qPCR and normalized to ACTIN. The comparative CT method (2^−ΔΔCT^) was applied to calculate relative differences in PCR results. (**E**) Western blot analysis of the MMP7 protein was performed on the total protein fraction of HEKa cells. The protein expression was normalized to the housekeeping protein ACTIN. The bars represent the mean ± S.D. (n = 6). Statistically significant variations: ** *p* < 0.01, *** *p* < 0.001 versus c-IgG treated cells.

**Table 1 ijms-24-11493-t001:** Patient classification.

	Samples	Age (Years)	Gender	aDsg3 (ELISA Score) ^a^	Clinical Phenotype of MPV
Control group (CTL)	1	45	F	/	
	2	53	F	/	
	3	59	M	/	
	4	59	M	/	
	5	61	M	/	
Study group (MPV)	6	59	F	+	Oral lesion
	7	50	M	++	Oral lesion
	8	36	F	+	Oral lesion
	9	51	F	++	Oral lesion
	10	67	M	++	Oral lesion

^a^ cut-off values are considered as 20 U/mL.

**Table 2 ijms-24-11493-t002:** Primers used for qPCR.

Gene	Accession Number	Forward (5′–3′)	Reverse (5′-3′)
*MMP7*	NM_002423.5	GGAGGCATGAGTGAGCTACA	TGCATCTCCTTGAGTTTGGC
*E-Cadherin*	NM_001317184.2	GAGCTGGACAGGGAGGATTT	AGTGTCCCTGTTCCAGTAGC
*TGF-β*	NM_000660.7	AAGATGGAGAGAGGACTGCG	AGAGGGAGAGAGAGGGAGTG
*ACTIN*	NM_001101.5	ACTCTTCCAGCCTTCCTTCC	CGTACAGGTCTTTGCGGATG

## Data Availability

The data presented in this study are available in the manuscript.

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
