# Peer review of "Plasma Exosomal microRNA Profile Reveals miRNA 148a-3p Downregulation in the Mucosal-Dominant Variant of Pemphigus Vulgaris"

_ijms, 2023, doi:10.3390/ijms241411493_

Round 1

Reviewer 1 Report

The article “Plasma Exosomal microRNA Profile Reveals miRNA 148a-3p Downregulation in Mucosal-Dominant Variant of Pemphigus vulgaris” is very interesting and I have some comments  with the intention of improving the manuscript

- Materials and methods:

- Did the patients have no cutaneous or other mucosal involvement?

- Some sentences like “The statistical significance of differential expression measured by miRNA sequencing was assessed by using the R package DESeq version 3.1.1 available in Bioconductor (Release 3.0). DESeq provides a method to identify differentially expressed (DE) miRNAs using the negative binomial distribution with variance. All p-values obtained were adjusted for false discovery rate.” You should be included in the material and method rather than in the results where you indicate this other phrase “All quantitative data were presented as the mean ± SD. Each experiment was performed at least six times. Statistical significance was evaluated using a t-test or one-way analysis of variance, followed by Bonferroni's test for multiple comparisons to determine statistical differences between groups. All the data were analyzed with the GraphPad Prism version 5.01 statistical software package"

Add when you consider a statistically significant difference.

- In the same way, there are phrases from the results that would be more appropriate to place in the discussion.

Otherwise the pilot study is very attractive for future diagnosis and treatment research of MPV

Thank you

Reviewer 2 Report

The comments to the authors:

In this study, the authors examined plasma exosomal microRNA (miRNA) in patients with mucosal-dominant-type of pemphigus vulgaris (MPV).  The authors found that miRNA 148a-3p was significantly down-regulated in MPV.  This is an important and novel study to understand the pathogenic role of plasma exosomal miRNA in MPV.  However, I have a number of comments, which are described below.

(1) The full spell for the abbreviation "EV" is not clearly mentioned.  "EV" may be extracellular vesicle.  However, the authors mention that the full spell "EV" is "plasma derived exosomes".  This point should be clarified at the first part of the manuscript, as well as throughout the manuscript .  The term "Dsg3-positive EV" should also be explained clearly both in the abstract section and in the first part of the text for the better understanding the readers.

(2) The terms "microRNA" and "miRNA" are inconsistently used in the abstract section.  This point should be improved.

(3) The exact name of the extracellular matrix, which is disrupted by metalloproteinse-7 should be described  at line 4 from the bottom in the abstract section.

(4) The abbreviations for desmoglein are not properly shown at line 2 in the introduction section.

(5) All the figures may be shown at the last part of the manuscript.  All the figure legends may also be shown at the last part of the manuscript in a separate page.

(6) "figure 6 D-E" at page 7, line 4 from the bottom should be corrected to "figure 5 D-E".

English needs sufficient improvements.  For example, the position of word "also” at line 7 in the abstract section may not be suitable.  The word "Metalloproteinse-7" at line 9 in the abstract section and at line 2 from the bottom of the introduction section should be changed to "metalloproteinse-7".  The term "EVs cargo" at line 14 in the abstract section should be changed to "EV cargo".  The comma (,) is not necessary at page 2, line 18.  The first characters in some words in the first line in the figure legend for the figure 2 should be changed to lower-case letters.  "Cadherin" should be changed to "cadherin" in the upper part of pate 6.  One of the two words for "used" should be removed at line from the bottom of the figure legend for the figure 4.  
